# Animals as Reservoir for Human Norovirus

**DOI:** 10.3390/v11050478

**Published:** 2019-05-25

**Authors:** Nele Villabruna, Marion P. G. Koopmans, Miranda de Graaf

**Affiliations:** Department of Viroscience, Erasmus MC, Wytemaweg 80, 3015 CN Rotterdam, The Netherlands; n.villabruna@erasmusmc.nl (N.V.); m.koopmans@erasmusmc.nl (M.P.G.K.)

**Keywords:** *Caliciviridae*, Norwalk, norovirus, host range, animal reservoir, pathogenesis, zoonosis, reverse zoonosis

## Abstract

Norovirus is the most common cause of non-bacterial gastroenteritis and is a burden worldwide. The increasing norovirus diversity is currently categorized into at least 10 genogroups which are further classified into more than 40 genotypes. In addition to humans, norovirus can infect a broad range of hosts including livestock, pets, and wild animals, e.g., marine mammals and bats. Little is known about norovirus infections in most non-human hosts, but the close genetic relatedness between some animal and human noroviruses coupled with lack of understanding where newly appearing human norovirus genotypes and variants are emerging from has led to the hypothesis that norovirus may not be host restricted and might be able to jump the species barrier. We have systematically reviewed the literature to describe the diversity, prevalence, and geographic distribution of noroviruses found in animals, and the pathology associated with infection. We further discuss the evidence that exists for or against interspecies transmission including surveillance data and data from in vitro and in vivo experiments.

## 1. Introduction

The majority of emerging infectious diseases that affect humans originate from animal reservoirs, predominantly wild life, including bats, rodents and birds. Norovirus is one of five genera of the family *Caliciviridae* and the most common non-bacterial cause of foodborne gastroenteritis worldwide. Noroviruses are currently categorized into at least seven genogroups (GI–GVII) that are further divided into more than 40 genotypes [1]. The virus contains three open reading frames (ORFs), ORF1 encoding the polyprotein that includes the viral polymerase, and ORF2 and ORF3 encoding the major- and minor capsid protein (VP1, VP2), respectively [2]. Recombination between ORF1 and ORF2 frequently occurs and therefore a dual nomenclature describing both the polymerase and capsid genotype is used [3,4,5]. Viruses from genogroups GI, GII and GIV are known to infect humans. Animal noroviruses including viruses found in pigs, dogs, and cats are closely related to human strains and cluster within GII (porcine norovirus) and GIV (feline and canine norovirus), respectively [1]. Noroviruses belonging to the other genogroups infect a broad range of hosts that includes livestock animals such as cows and sheep but also marine mammals and rodents. In the past years, an increasing number of metagenomic studies have led to the discovery of additional noroviruses in new animal hosts and it seems evident that we lack understanding of the full diversity of noroviruses and their host range [6,7]. Most human infections and outbreaks are caused by viruses belonging to GI and GII. The GII.4 genotype viruses have been particularly prevalent in the past two decades, and evolve through accumulation of mutations but also by recombination. Such recombinants and other new genotypes emerge regularly but the origin of these new viruses is not well understood [8]. This regular detection of novel strains and the reporting of human-like norovirus genotypes in stool samples of symptomatic and asymptomatic farm animals have sparked interest in the possible role of animals as potential zoonotic reservoir for these emerging strains [9,10,11,12]. Antibodies directed against bovine and canine norovirus have been detected in humans suggesting some level of exposure of humans to animal norovirus [13,14,15,16]. For other viruses of the *Caliciviridae* family, interspecies transmission has been reported including some case reports of zoonotic events between marine mammals and humans (reviewed in [17]).

This systematic review summarizes the literature on the known animal reservoir for norovirus, the virus diversity, prevalence, and geographic distribution, as well as pathological findings associated with norovirus infections in animals. We will further discuss the existing evidence and probability of interspecies transmission including susceptibility of animals used as models in norovirus research. There are several reviews that focus exclusively on the role of mice in norovirus research [18,19,20]; therefore, we will discuss murine norovirus only in context of surveillance of wild animals. Molluscs are an important vehicle of foodborne norovirus transmission, but do not support norovirus replication and have been reviewed elsewhere [21,22].

## 2. Results

### 2.1. Search Output:

The search yielded 6702 papers of which a total of 182 were included in the review. An additional nine papers were later included (see methods).

### 2.2. Noroviruses in Domesticated and Wild Animals

Norovirus was first described from a gastroenteritis outbreak in 1968, which affected children in a school in Norwalk, Ohio, USA [23]. In 1972, the virus was visualized for the first time by immune electron microscopy revealing “small round structured viruses” (SRSV) of 27–35 nm in diameter, which was used as their first classification [24]. Viruses of similar morphology were soon described from stool samples of domestic calves and pigs, and sequencing confirmed the presence of viruses belonging to the same family as human noroviruses. To date, porcine noroviruses are genetically most similar to human norovirus; porcine noroviruses have been classified among a diverse range of human norovirus genotypes in GII as GII.11 (prototype SW918), GII.18 and GII.19 [10,25] and have been found in stools and intestinal content of pigs all over Europe, North and South America, and Asia (Figure 1A,B, Table 1).

In most countries, the overall detection rate of porcine norovirus in stool samples is low (0–16.6%) and outbreaks have not been reported, although there is evidence for symptomatic porcine norovirus infections. When specific-pathogen-free (SPF) piglets were inoculated with GII.11 or GII.18 positive fecal filtrate they showed mild to moderate diarrhea within 1 day post inoculation (dpi) and norovirus RNA was amplified from intestinal content as well as from sera [26,27]. The majority of surveillance studies have been screening healthy pigs from farms and slaughterhouses [9,10,25,28,29,30,31,32,33,34,35,36,37,38,39,40,41,42,43,44]. Asymptomatic finisher pigs most commonly tested positive, but porcine noroviruses have also been found in stools from asymptomatic pigs of other age categories as well as diarrheic piglets [26,34,45]. Virus circulation is thought to be widespread. A survey of pigs found antibodies to GII.11 virus like particles (VLPs) in 71% and 36% of pigs in the USA and Japan [46].

The SRSV found in stool samples from cattle have subsequently been characterized as bovine norovirus GIII.1 (Jena agent) and GIII.2 (Newbury Agent 2), discovered in cattle in Germany and England, respectively [12,118]. Upon experimental inoculation with a GIII.1 or GIII.2 gnotobiotic calves develop diarrhea, shed virus for several days and seroconvert, although not in 100% of the cases [11,103,109,119,120,121,122,123,124]. Both genotypes are widely distributed among diarrheic and healthy cattle, juveniles, and adults, although GIII.2 viruses have been found more frequently than GIII.1. The majority of published surveys has tested diarrheic calves, in which bovine norovirus was frequently found [13,42,50,51,63,69,71,72,78,79,91,92,100,102,109,115,116]. One case–control study that investigated pathogens associated with calf diarrhea in the USA tested 444 samples of 1–2 week-old diarrheic and asymptomatic calves for a panel of 11 enteric pathogens (bacteria and viruses) using real-time RT-PCR with bovine norovirus specific probes. A prevalence of 44.7% was reported in diarrheic and 16.3% in healthy calves [50]. Less is known about bovine norovirus in adult cattle. One study compared prevalence of bovine norovirus RNA in pooled manure samples of 75 dairy farms with those of 43 veal calf farms. A high proportion (44%) from the veal calf farms was positive, but bovine norovirus RNA was not detected in samples from the dairy farms [9]. The prevalence of antibodies to GIII.1 or GIII.2 VLPs was >70%, independent of location (Table 1) and only very few studies failed to detect GIII viral RNA or antibodies (Figure 1B). A proposed third GIII genotype, GIII.3, was found in asymptomatic sheep in New Zealand [38].

While pigs and cows are the best studied non-human hosts—apart from mice—noroviruses have also been detected in stool samples from cats and dogs. Both animal species were shown to be infected by viruses belonging to genotype GIV.2, while dogs are also hosts of canine GVI and GVII strains. The first carnivore norovirus was documented in a captive lion cub (*Panthera leo*) in Italy that had died of severe hemorrhagic enteritis [84]. This new strain shared ~70% aa VP1 identity with the human GIV.1 sequence, which is only identified sporadically in the human population, but is more commonly detected in sewage samples [125]. One outbreak study documented the arrival of two diarrheic young dogs into a kennel in Lisbon [88]. Two days later, five young dogs housed in the same kennel developed diarrhea and the isolated GVI.2 sequences were identical to each other. Canine noroviruses sequences have since been detected in feces from healthy and sick dogs from kennels, shelters, and households in South America, Europe, and Asia (Figure 1, Table 1). To date, no infection studies have been conducted with canines and the pathology of noroviruses in dogs is therefore unclear. However, during a study in Portugal, canine norovirus RNA was found more often in the stool samples of symptomatic dogs compared to asymptomatic dogs (40% versus 9%), suggesting they play an important role as cause of disease [87,126]. In a Europe-wide study, an overall 4.4% prevalence was found for diarrheic dogs while none of the healthy animals tested positive [83]. A strong seasonal pattern was observed during a four year period of sampling dogs in Portugal, with the highest prevalence (36%) in winter and lowest (7%) in autumn, similar to the seasonality observed for norovirus in humans [89,126]. A serological survey screening dogs from 14 different countries found variable prevalences of antibodies to GVI.2, ranging from 0% in Hungary and Ireland up to 60% in Portugal [47].

The first evidence for feline noroviruses was provided through an Italian study, where 16% of cats tested positive for GIV.2-specific antibodies, with the highest prevalence among stray cats (32%) [81]. Three years later, in 2012, a feline norovirus was discovered during a gastroenteritis outbreak in cats in a shelter in the USA [53]. The cats were negative for known feline parasites, but a full norovirus genome was recovered (JF781268). Similar viruses were later detected, mostly in diarrheic cats [54,62,65,85]. After inoculation of SPF cats with feline norovirus, the cats shed the virus up to 7 dpi, viral RNA was detected in sera of all cats, three of the four cats developed diarrhea and one started vomiting [127]. Another study using the same inoculum showed that cats developed IgG against recombinant VP1 protein identical to the strain used for the experimental infections [128].

Apart from domesticated animals, noroviruses have also been detected in wild animals, such as harbor porpoises (*Phocoena phocoena*) and californian sea lions (*Zalophus californianus*) [55,98]. Neither of these viruses could be assigned to an existing genogroup. Further investigation found 10% of harbor porpoise intestinal tissues RT-PCR positive and 24% of the animals seropositive for porpoise norovirus, suggesting that norovirus infections are common in these animals. With the recently increasing trend of metagenomic studies, additional norovirus have been identified. In a metagenomics analysis of bats intended to decipher the bat virome, a full norovirus genome was recovered from intestinal tissue of *Rhinolophus pusillus* bats captured in two Chinese provinces [6]. In one location the prevalence in fecal samples was as high as 20%. This strain belongs to a new genotype which shares highest sequence homology with GV norovirus (Figure 2) [129]. Subsequent studies have detected norovirus in two species of insectivorous bats in China, namely *Rhinolophus sinicus* and *Rhinolophus affinis* [64,130]. Most of the animal noroviruses have not been detected in animals other than the species were they were first identified in. Exceptions are the GV noroviruses, which are detected in mice and rats, and the canine/feline GIV and GVI noroviruses.

### 2.3. Is There Evidence for Cross Species Transmission?

Since the first norovirus has been detected from animals, the question has been raised whether norovirus can jump the species barrier. To date, there are no controlled outbreak studies during which both animals and humans have been sampled simultaneously. One calicivirus outbreak in a nursing home in 1983 in the UK was epidemiologically linked to a sick dog. While virus particles were found in the patients, no stool sample was available from the dog and only antibodies against the same virus could be detected [131].

#### 2.3.1. Animal-to-Human Transmission

To date, no animal norovirus have been detected in human stool, but some serological evidence hints to possible transmission from animals to humans. This includes a handful of studies that reported seroprevalence against bovine [13,14,132] and canine [15,16] norovirus in humans. A Dutch study compared antibody titres against GIII.2 VLPs from 210 bovine or porcine veterinary specialists against age, sex, and residence matched controls with the aim to evaluate whether higher exposure to animals is reflected in increased titers against animal noroviruses [132]. More veterinarians had anti-GIII.2 IgG antibodies compared to the control group (28% versus 20%). Similarly, the seroprevalence of antibodies to canine GVI.2 VLPs was tested in a cohort of 373 veterinarians versus age, sex, and district matched controls. Of the veterinarians, 22.3% were seropositive for GVI.2 in comparison to 5.8% in the control group [15]. Anti-GIII antibodies were also detected in 26.7% of adult blood donors in Sweden [14] and in a birth cohort in India, which compared seroprevalence of mothers and their children [13]. However, the possible presence of cross-reactive antibodies needs to be considered in these studies: the GIII.2 response was in part correlated with GI.1 response, but not with the GII.4 response. The finding that some sera contained higher antibody titers against GIII.2 than human norovirus indicates that not all anti-GIII.2 response can be explained by cross-reactivity [132]. Importantly, no cross-reactivity between bovine GIII.2 and human GI.3, GII.1, GII.3, GII.4, GII.6 was detected when convalescent anti-GIII.2 sera of a gnotobiotic calf or specific anti-GIII.2 or GII.3 antibodies were used [14,121]. Cross-reactivity between GVI.2 and GII.4 was assessed by pre-incubating GVI.2 positive sera with GVI.2 VLPs before assessing their binding to GII.4 or GVI.2. Preincubation with GVI.2 blocked binding to GVI.2 VLPs but had no effect on sera binding to GII.4, suggesting that these two genotypes share no conserved epitopes [15]. In contrast, cross-reactivity was observed between more closely related human GIV.1 and canine GIV.2 noroviruses in an age stratified cohort of 535 people in Italy [16], where 28.2% of the sera reacted to both GIV.1 and GIV.2 VLPs and only 0.9% detected exclusively GIV.2 VLPs.

#### 2.3.2. Human-to-Animal Transmission

Numerous studies have investigated the possibility of human norovirus transmission to animals by screening animal stool samples for human noroviruses or by investigating the seroprevalence against human norovirus strains (Figure 3, Appendix A). The closest to an outbreak study was one case-control study that included 92 dogs from Finnish households. The main inclusion criterion was that either the dog or a human in the household had suffered from vomiting or diarrhea [110]. Four dogs tested PCR positive and they all came from households in which at least two people suffered from severe gastroenteritis symptoms that had disappeared not longer than three days before the dog samples were taken. Based on a ~370 nt region two GII.4 variants and one GII.12 genotype were identified, of which one GII.4 was identical to the virus found in the owner’s feces. The other strains were >98% nt identical to circulating human norovirus strains. Antibodies against GII.4 and GI.1 VLPs have been detected in dogs sampled in a European study and against GII.4 and GIV.1 in dogs in Italy [47,80]. Both studies found that sera from some animals reacted exclusively to the human strains but not to canine GVI.2 VLPs. Caddy et al. investigated the seroprevalence against human noroviruses (GI.1, GI.2, GI.3, GII.3, GII.4, GII.6, GII.12) in two dog populations; sera from dogs in a rehoming kennel in 1999–2001 and sera collected in 2012–2013 from a diagnostic lab. Overall, seropositivity against GI was very low, but 10.7–18.6% were seropositive against GII VLPs [106]. The majority of seropositive dogs had antibodies detecting GII.4 viruses which was the most prevalent human norovirus circulating during this time. Only weak cross-reactivity was observed with canine sera or polyclonal sera specific for GII.4 or GVI.1/GIV.2 [106]. Combined, these studies suggest that human noroviruses could infect dogs, although more work is needed to unravel potential cross-reactivity with non-human viruses, like GVI.2 [80].

Several surveys in pigs reported human norovirus in pig feces and two reported more than one genotype [30,34,36,43,117]. In a longitudinal study in Japan intestinal content of 20 apparently healthy 6 month-old pigs were screened each month with calicivirus-specific primers. Of these, 11/354 were positive for human GII without a seasonal pattern being recognized [36]. Based on partial capsid sequences these strains were classified as GII.4, GII.3 and one GII.13, all three genotypes that had been reported in outbreaks in humans during that season. Another study tested 530 fecal samples of asymptomatic pigs (<8 month) from six farms in Taiwan, 7% tested positive with RdRp-specific primers, while GII capsid specific primers resulted in 32% positive samples, 41% in winter and 26% in summer [34]. The GII.4 and GII.2 classified sequences were found in pigs of all age categories and from different farms. Sequences of GII.1 and GII.4 noroviruses have also been detected in feces of two healthy sows in Ethiopia and GII.4 in pig feces from two different farms in Canada [30,43].

Antibodies recognizing human norovirus have been detected in healthy household pigs in Nicaragua and US pigs with prevalences ranging from 52%–70% [46,133]. While those antibodies recognized VLPs of GI.1, GII.1, GII.3 and GII.4 they were not able to block their binding to pig mucin [133]. Cross-reactivity was also investigated and antibodies against GII.1 and GII.3 but not against GI genotypes cross reacted with porcine GII.11 [46]. The studies thus far raise the question if certain norovirus genotypes considered to be “human” noroviruses co-circulate among pigs. As these observations are not consistent, this could be restricted to some regions where opportunity for contact of pigs with humans is higher.

During the 2014–2015 epidemic season, GII.17 was the dominant human norovirus genotype in some Asian countries [134,135]. 32 of 50 rhesus macaques on a Chinese farm tested positive using GII.17 specific primers and a whole GII.17 genome (KX356908) was recovered from one animal [136,137]. This GII.17 genotype was 99% identical to a human GII.17 recently detected in China [137]. Rectal swabs of juvenile rhesus macaques from a primate research center in the USA were screened by real-time RT-PCR for GI, GII, and GIV noroviruses; of the 500 samples, 8.2% were positive [138,139]. Sanger sequencing showed that the animals were positive for 30 GI.1 and eight GII.7 strains, and yielded two full ORFs of GI.1 and GII.7 sequences (KT943503–KT943505). Surprisingly, the GI.1 sequences were not only identical to each other but also to the prototype Norwalk virus described in 1968. The GII.7 sequences were 99–100% identical to each other and 95% identical to a human norovirus (KJ196295). Furthermore, antibodies against various human norovirus genotypes were detected in captive primates in the US; IgG against GI.1, GII.4, GII.5 and GII.7 VLPs were detected in mangabeys (85%), macaques (~60–65%), and chimpanzees (92%) [140,141]. 

Compared to surveillance in livestock animals only a few studies have investigated wild animals. Bird feces were collected during three winters (2009–2011) from fresh snow of a household waste dumping site in Finland and analyzed by GI and GII specific real-time RT-PCR [111]. Of the 115 avian feces tested, six were positive for GI and 25 for GII, albeit with high Ct values, the lowest being 36. Sequencing and typing was successful for four GII.4 (GII.4 2006a/b, 2009) and two GII.3 viruses, all at least 94% identical to known human strains. Based on cytrochrome c oxidase I sequencing, the positive feces could be assigned to gulls and crows. A human norovirus was found in the intestinal content of a dead Norway rat that had been trapped in the sewer system in Copenhagen; a ~4000 bp sequence was recovered and was typed as a GI.Pb-GI.6 strain [108]. The virus titer was calculated to be 5 × 10^7^ genome copies/g feces and norovirus particles were detected in feces by immunogold electron microscopy [108]. 

#### 2.3.3. Susceptibility of Animals to Human Norovirus Strains

In addition to finding human norovirus in animal stool samples, noroviruses have been found to cross the species barrier under experimental conditions. To date, seven animal models have been developed to study human norovirus infection; gnotobiotic calves and pigs, immunocompromised BALB/c Rag-^γc^-deficient mice, Yucatan miniature pig, and three non-human primates, namely chimpanzees, rhesus, and pigtail macaques (Table 2). In contrast, common marmosets, cotton top tamarin, immunocompromised ferrets, and cynomolgus macaques were not found to be susceptible to infection, although only a limited number of norovirus genotypes was tested [142,143]. All models support viral replication evident by viral shedding and seroconversion upon oral or intragastric inoculation with a high viral dose (10^4^–10^6^ genomes). Whereas pigs and calves developed diarrhea, both chimpanzees and rhesus macaques did not display any gastrointestinal symptoms. Virus replication was usually found to be restricted to sites of the small intestine. In mice, viral genomes could be amplified from various organs, and in minipiglets, low levels of the virus were additionally found in blood as well as in tonsils, spleen, and lymph nodes [144,145]. Pathological changes were detectable only in calves and pigs but not in primates. These changes included villous blunting, atrophy, and an increase in inflammatory cells in the lamina propria. Norovirus antigen was detected in the small intestine, varying between duodenum, jejunum, and ileum depending on the animal and the virus strain used for inoculation. Noteworthy, in pig as well as in chimpanzee experiments, animals were chosen based on their histo-blood group antigen (HBGA) and secretor status. In pigs, take of infection was strongly dependent on their HBGA phenotype and secretor status. HBGA type A+/H+ pigs were more readily infected than type A-/H- pigs [146]. However, while two culturing systems have been described for human norovirus [147,148], attempts to grow human norovirus in animal cell culture have not yet been successful [149,150].

The best understood host factors influencing susceptibility to human norovirus infections are the HBGA, glycans that act as attachment factors for norovirus, and the host secretor status [165,166,167,168,169,170]. Alternative attachment factors, including sialic acids and heparan sulfate, have been proposed and it is likely that other cell surface molecules play a role in norovirus binding to the cell [171,172,173,174]. Virus attachment is a prerequisite for a cell’s susceptibility to infection and studying a host’s or population’s HBGA distribution can imply putative target cells and susceptible populations, respectively; HBGA expression and distribution within a host can indicate virus cell tropism while their expression in different putative human and animal hosts can be an indicator for host range.

A host’s HBGAs type is determined by the ABO- and Lewis blood group systems. ABO synthesis begins with the addition of fucose to a carbohydrate precursor on glycoprotein or glycolipid precursor structures by a α1-2 fucosyltransferase. This enzyme is expressed from two separate loci (*H* and *Se*) one expressed on red blood cell precursors, the other on epithelia cells of the gastrointestinal, respiratory, and reproductive tract. Individuals who have a non-functional fucosyltransferase 2 (*FUT2*) version express the H antigen only on their red blood cells but not in their gastrointestinal tract. The A and B antigen are subsequently added onto the H antigen by various other glycosyltransferases. Lewis antigens are sugar moieties, consisting of a precursor structure, or the A, B, H antigens to which an extra fucose group has been added. The *Se* locus also determines whether soluble forms of the ABH antigens are secreted into bodily fluids. Humans with an inactive *Se* gene are referred to as non-secretors since no ABH antigens are found in their saliva and mucus [175]. Noroviruses bind these HBGA in a strain dependent manner, thus leaving non-secretors non-susceptible to some norovirus strains. In pigs and primates, the HBGA phenotype seems to be important for infection with human norovirus as well. In animal studies the host’s HBGA phenotype and virus strain used for inoculation can be selectively paired. Binding assays have been used as an alternative surrogate to study interaction between virus attachment factors (Figure 4, Appendix A). 

Animal or human norovirus VLPs or purified virus can be tested with regards to their attachment to either animal or human saliva or tissue with known HBGA content. Canine and the newly discovered norovirus from bats appear to attach to HBGAs similar to human noroviruses [129,176]. Bovine GIII.2 and murine GV have been shown to be dependent on receptors that are not thought to be expressed in humans; GIII.2 strains do not bind the same sugar moieties as human norovirus but to a αGal 1–3 sugar (Gala3Galb4GlcNAcb-R) instead [177]. This epitope is expressed in all mammals with the exception of the *Hominidae* family. In line with this, GIII.2 particles bound strongly to bovine saliva but neither to human saliva nor duodenal tissue. 

GV infection in mice was reported to depend on terminal sialic acids and glycoproteins on macrophages, in a strain dependent manner [178]. Recently, a proteinaceous receptor, CD300lf, was detected in mice, which is expressed on tuft cells that are present in small numbers in the intestine as well as on cells of the hematopoietic/myeloid lineage. However, the human CD300lf homologue does not function as receptor for human or murine norovirus [74,179]. For other noroviruses, including porcine and feline genotypes, no attachment factor or receptor is known.

Most of the susceptible hosts mentioned above, with the exception of several fish and bird species, contain a *FUT1* and *FUT2* gene. The lack of these genes can be potentially compensated for by another fucosyltransferase, or alternatively in these newly discovered animal norovirus, could attach to an alternative receptor [180].

## 3. Discussion and Conclusions

More than two thirds of emerging infectious diseases that affect humans originate from animal reservoirs, predominantly bats, rodents, birds, and other wildlife, and therefore, we sought to review evidence for interspecies transmission of noroviruses [181]. While most of our understanding about the norovirus animal reservoir stems from domestic animals, the recently increasing number of metagenomic studies, investigating the virome in a more unbiased way, have extended the norovirus host range by new species, while simultaneously complementing the knowledge about norovirus diversity. For many of these newly discovered viruses, we have little more information than a genome sequence and it remains to be determined if they indeed are host specific. Bats, wild rodents, and birds are known to frequently host pathogens that can cause disease, but have hardly been studied for evidence of norovirus infection. 

Our review found more evidence for human noroviruses in animals than the reverse, suggesting that human norovirus could be a reverse zoonosis, with identification of human norovirus RNA in stool samples from pets, rodents, birds, pigs, and cattle. However, the question is what constitutes evidence for infection, as it can be argued that the detection of norovirus in feces indicates ingestion of norovirus contaminated material rather than an active infection. The molecular RNA detection methods can be sensitive enough to detect amounts as low as 10 virus genomes and such low virus levels could be due to ingestion [182]. To establish that both species can serve as a host, detection of either replicating virus by increase in virus titer over time, a specific immune response, or detection of proteins that are only expressed upon infection is required. This has been shown experimentally in cattle, pigs, macaques, and chimpanzees, confirmed by seroconversion and virus shedding. Serological studies can also be used to confirm viral detection in field studies, thus increasing the window of detection, as antibodies persist much longer than virus shedding. However, serological assays have their draw backs: antibodies can potentially also be induced by exposure to the virus rather than infection and cross-reactivity has to be taken into consideration when analyzing the results. Cross-reactivity has been described primarily between strains within one genogroup and less between viruses from separate genogroups [183]. This is of importance when analyzing serology data against human and animal noroviruses that cluster in the same genogroups, such as porcine, feline, and canine noroviruses. Many serology studies reported some sera that contained antibodies only recognizing animal strains but not humans or vice versa, increasing the chance that these are specific antibody responses. Serology has the advantage of providing information about the prevalence of a pathogen in a certain host species without relying on samples to be taken during an active infection. It is therefore a good tool to screen potential hosts with regards to their risk of exposure. However, this data should be complemented by detection of viable virus from the host. Since culturing is difficult for norovirus, deep sequencing to detect viral genomes is for now the best alternative. Should human norovirus infect animals the question remains whether these interspecies transmissions are relevant for human infections; if once transmitted to animals, these strains can be re-introduced into humans. Furthermore, strains that only cause sporadic infection in humans, such as GIV noroviruses, could reside in an animal reservoir between outbreaks. 

Evidence for transmission of animal norovirus to humans is sparse and solely based on serological evidence. If these transmissions occur they are likely to be rare events that could be difficult to detect if they are asymptomatic or sporadic infections. In addition surveillance is not developed to detect these viruses in human stool samples. Several papers reported differences in detection rate based on their choice of primers; protocols with GI or GII specific probes will potentially miss the animal noroviruses, while the generic calici- or norovirus primers that are often used for detection of human and animal noroviruses in animals might have lower sensitivity compared to more specific primers [34,53,77,89,99,139]. It is open to debate whether some viruses that are categorized as human norovirus today might have originated from an animal source; the origin of newly emerging recombinants, such as the GII.pe polymerase, is unknown and it is a possible scenario that these new recombinants are the result of a recombination event between an animal and a human norovirus. Recombination occurs primarily within genogroups and only three intergenogroup recombinants namely between GI.3–GII.4, GII(NA)–GVI, and feline GIV.2–GVI.1, have been identified [127,184,185]. Recombinants are also found within bovine, porcine, canine, and feline genotypes. The formation of human-animal norovirus recombinants is a possible scenario, especially for animal genotypes that cluster close together with human genotypes. Water, food sources, and filter feeding shellfish can harbor a variety of multiple human and animal genotypes and genogroups simultaneously thereby posing a possible source of co-infection in humans and animals [186,187,188,189]. Based on the current body of evidence it is too early to consider norovirus a zoonotic or reverse zoonotic pathogen. To increase chances of catching a transspecies transmission event more targeted surveillance would be needed; to include samples of animals and humans that are in close contact, ideally during an outbreak situation and with an unbiased detection method [15,131,132,190]. Unravelling norovirus reservoirs and movement between species will help us understand norovirus evolution and emergence.

## 4. Methods

### 4.1. Search Strategy

We searched the literature in the Embase, Medline Ovid, Web of science, and Google scholar databases, using the search strings shown below. Number of papers found is depicted in brackets.

#### 4.1.1. embase.com (2903)

(“Norovirus”/exp OR “norovirus infection”/exp OR (Norovirus* OR Norwalk OR “small round-structur*” OR srsv*):ab,ti) AND ([animals]/lim OR “reservoir”/exp OR (nonhuman/de NOT human/exp) OR “zoonosis”/de OR “disease model”/de OR (animal* OR reservoir* OR nonhuman* OR non-human* OR animal* OR rat OR rats OR mouse OR mice OR murine OR dog OR dogs OR canine OR cat OR cats OR feline OR rabbit OR cow OR cows OR bovine OR rodent* OR sheep OR ovine OR pig OR swine OR porcine OR veterinar* OR chick* OR baboon* OR nonhuman* OR primate* OR cattle* OR goose OR geese OR duck OR macaque* OR avian* OR bird* OR mammal* OR poultry OR bat OR porpoise* OR zoono* OR farm OR farms OR “disease model*”):ab,ti)

#### 4.1.2. Medline Ovid (1550)

(Norovirus/OR (Norovirus* OR Norwalk OR small round-structur* OR srsv*).ab,ti.) AND ((exp animals/NOT exp humans/) OR Disease Reservoirs/OR Zoonoses/OR Models, Animal/OR Disease Models, Animal/OR (animal* OR reservoir* OR nonhuman* OR non-human* OR animal* OR rat OR rats OR mouse OR mice OR murine OR dog OR dogs OR canine OR cat OR cats OR feline OR rabbit OR cow OR cows OR bovine OR rodent* OR sheep OR ovine OR pig OR swine OR porcine OR veterinar* OR chick* OR baboon* OR nonhuman* OR primate* OR cattle* OR goose OR geese OR duck OR macaque* OR avian* OR bird* OR mammal* OR poultry OR bat OR porpoise* OR zoono* OR farm OR farms OR disease model*).ab,ti.)

#### 4.1.3. Web of Science (2049)

TS = (((Norovirus* OR Norwalk OR “small round-structur*” OR srsv*)) AND ((animal* OR reservoir* OR nonhuman* OR non-human* OR animal* OR rat OR rats OR mouse OR mice OR murine OR dog OR dogs OR canine OR cat OR cats OR feline OR rabbit OR cow OR cows OR bovine OR rodent* OR sheep OR ovine OR pig OR swine OR porcine OR veterinar* OR chick* OR baboon* OR nonhuman* OR primate* OR cattle* OR goose OR geese OR duck OR macaque* OR avian* OR bird* OR mammal* OR poultry OR bat OR porpoise* OR zoono* OR farm OR farms OR “disease model*”)))

#### 4.1.4. Google Scholar (200)

Norovirus|Norovirusses|Norwalk|“smallround-structur”|srsv animal|animals|reservoir|nonhuman|zoonosis|zoonoses|"disease model"

### 4.2. Selection Criteria

Two independent reviewers screened titles and abstracts for their relevance. We included publications that mentioned norovirus in the title or abstract but we excluded papers about food (oyster) and waterborne outbreaks, food surveillance or food related experiments, and oyster/seafood surveillance. We excluded papers on murine noroviruses as models. Papers describing norovirus surveillance in wild mice and papers using mice as model for human norovirus were included (Figure 5).

In a second round, we screened the papers for whether they described (1) animal surveillance studies to detect human or animal norovirus by PCR, sequencing or by serosurveillance including negative results; (2) experimental animal infections with human or animal norovirus; (3) human surveillance studies to detect animal norovirus by PCR, sequencing or by serosurveillance including negative results; (4) animal norovirus characterization including molecular assays and genome announcements.

### 4.3. Data Extraction

Of the remaining papers, the following data was extracted:General description.Location (country, district, city), duration of study, date of study, species and number of tested animals and age of animals. For studies describing experimental infections of animals with human or animal noroviruses, the following information was collected if described in the paper: Details on experimental infection methods.Regarding the experimental infection, the route of inoculation was documented since this may affect which subclasses of immunoglobulins are induced. In addition, genogroup/genotype of the virus inoculate, as well as amount used (number of genome copies) and the sample type collected (e.g., saliva, feces, sera) were registered. It was further recorded how virus replication was confirmed, which methods was used to detect virus (RT-PCR, real-time RT-PCR, antigen capturing ELISA, EM), how much was detected and at what time points. Details on clinical picture; description of the health state of the animals; which symptoms (e.g., diarrhea, vomiting), as well as the duration of symptoms.Pathology; pathological examination results.Immunohistochemistry data was extracted to for information regarding the organ and cell tropism.Host response was assessed by collecting serological data including method of antibody detection, type of immunoglobulins (Igs) tested (IgM, IgG, IgA), origin of Igs (saliva, sera, feces), the time period Igs were detected and if available whether they were blocking virus from binding to HBGAs. Since some animal noroviruses cluster close to human norovirus, information about cross-reactivity was also collected. Host factors such as HBGA, secretor and non-secretor status were of interest, since they are known to be important for susceptibility in humans, while in animals this link is less evident.

For surveillance studies additional data was collected regarding duration of surveillance, species, setting of the animals (farm, slaughterhouse, research facility, households, and the wild), and type of farm (if applicable; indoor/outdoor/free range). When virus shedding was detected by RT-PCR, it was noted which region of the genome was detected and whether the ORF1/ORF2 overlap was amplified. Furthermore, the similarity of new virus sequences with known sequences in the database was recorded. When sequences were available, they were re-typed with the Noronet typing tool.

## Figures and Tables

**Figure 1 viruses-11-00478-f001:**
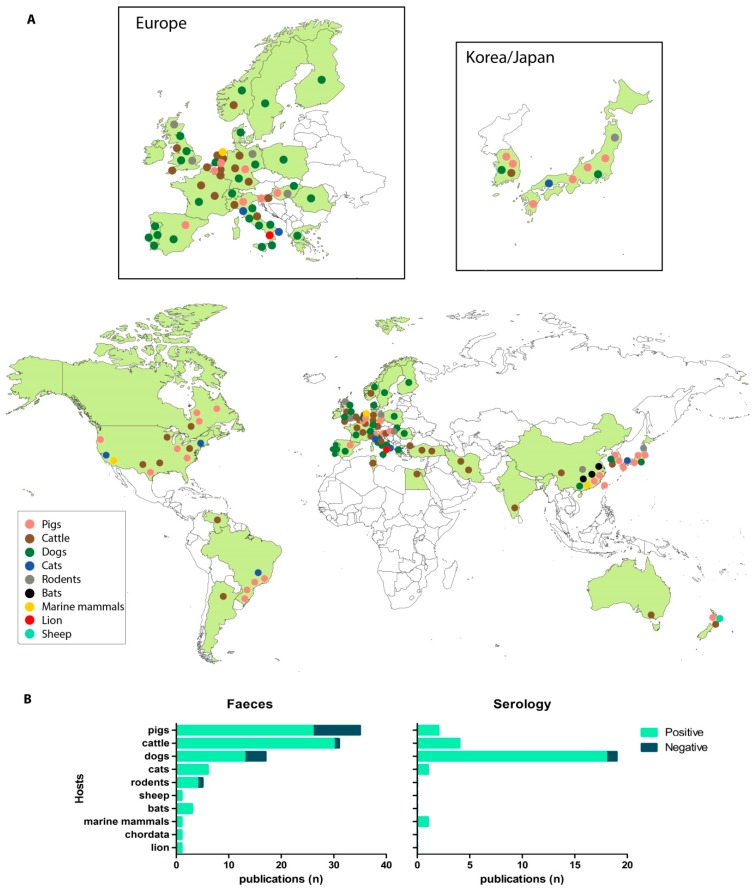
Studies describing the presence of animal norovirus across the world. (**A**) Countries in which animal norovirus have been detected are colored green. Each dot represents a study and location where animals have been found positive by either RT-PCR, real-time RT-PCR, or serology. The color indicates the host. (**B**) Number (n) of publications reporting positive versus negative surveillance results in different hosts for PCR results in feces and serology studies. Note that a paper that studied GVI.2 seropositivity in dogs in 14 European countries is listed as 14 studies in 1B [47]. Details of the studies are listed in Table 1 and Appendix A.

**Figure 2 viruses-11-00478-f002:**
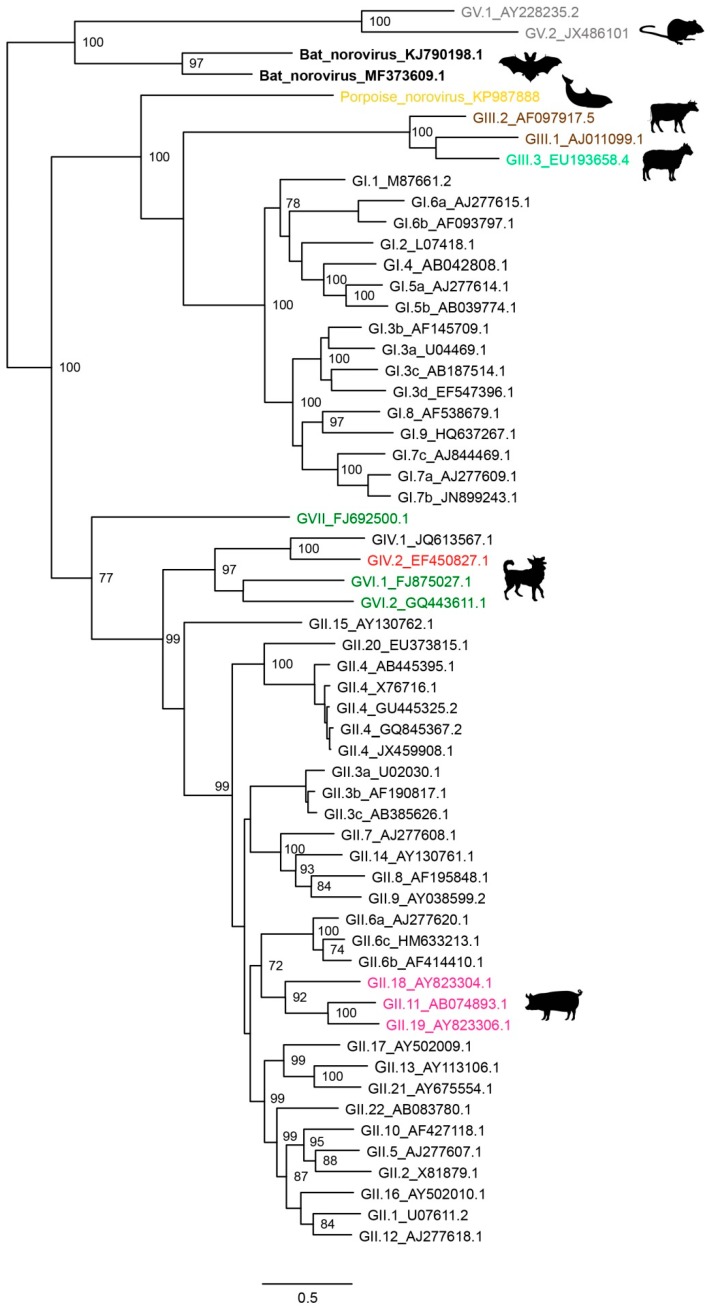
Maximum-likelihood tree of open reading frame 2 (ORF2). The tree was inferred by PhyML 3.0 software (http://www.atgc-montpellier.fr/phyml/) by using the general time reversible nucleotide substitution model. Bootstrap values >70 are shown. Scale bars indicate nucleotide substitutions per site. Animal noroviruses are colored with same color code as in Figure 1A.

**Figure 3 viruses-11-00478-f003:**
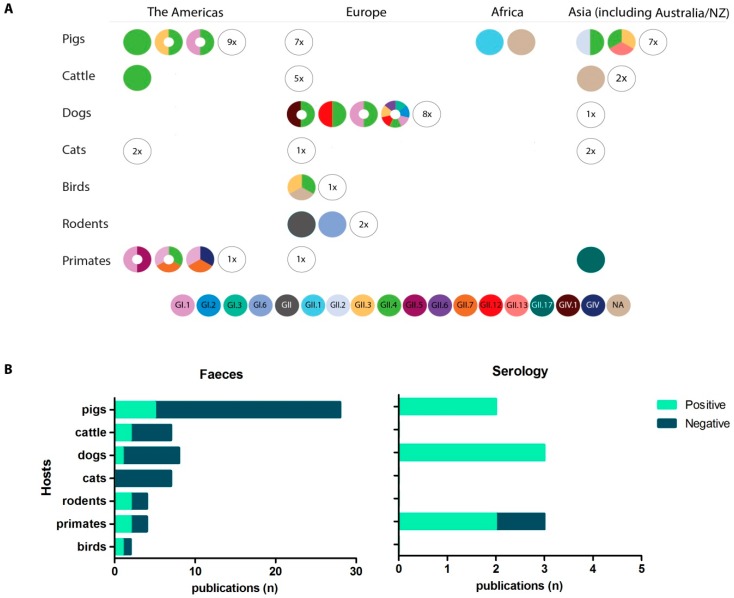
Human norovirus genogroups and genotypes detected in studies investigating human-to-animal transmission. (**A**) Studies that analyzed fecal samples for human norovirus sequences by RT-PCR, real-time RT-PCR or serological studies. Every circle represents one study and colors represent different norovirus strains identified through sequencing. Serological studies are marked with a central white circle, and colors here represent antigens used for the serological testing. Numbers in empty circles indicate the number of studies in which no evidence for human norovirus infection was found. NA stands for studies where the genogroup or genotype was not identified. (**B**) Number (n) of virological and serological studies of norovirus in different hosts, grouped according to results (positive versus negative). More details can be found in Appendix A. NZ = New Zealand.

**Figure 4 viruses-11-00478-f004:**
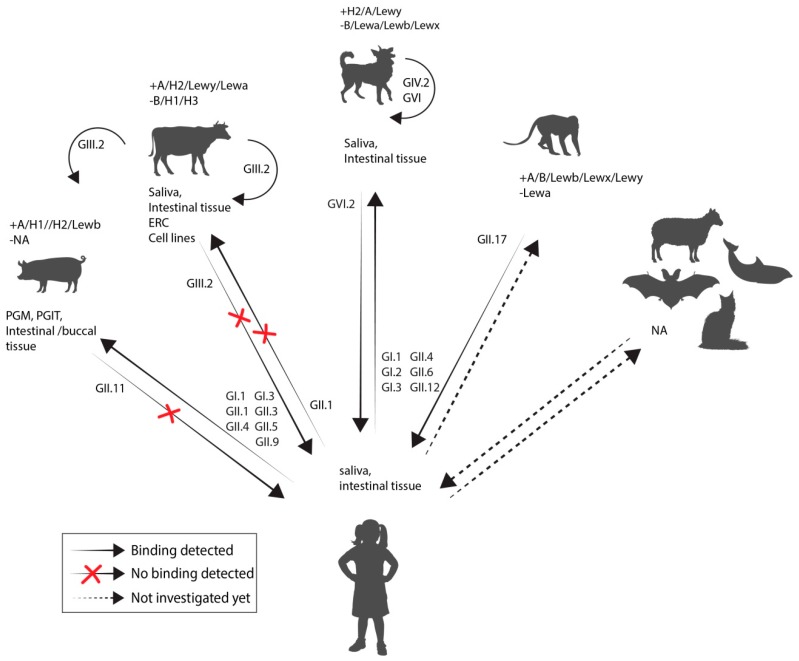
Results of binding studies with animal and human norovirus VLPs. The histo-blood group antigen (HBGA) phenotype is indicated with the presence (+) or absence (−) for different glycans. Arrows indicate direction in which attachment was tested and whether attachment was observed or not (red cross). Dotted arrows indicate that attachment has not been assessed yet. The half circular arrows indicate binding of animal norovirus to tissue/saliva of either the same or another animal species. Detailed information about the individual studies can be found in Appendix A.

**Figure 5 viruses-11-00478-f005:**
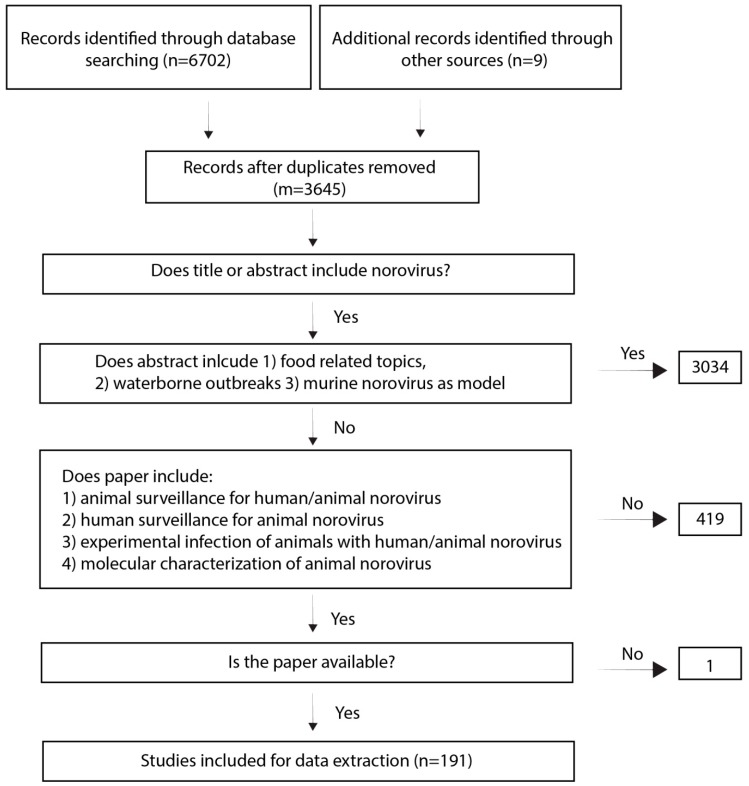
Inclusion and exclusion criteria for paper selection.

**Table 1 viruses-11-00478-t001:** Summary of studies detecting animal norovirus in animals, either in feces or by serology. Details of each study can found in Appendix A.

	Location	Host	Genotype	Prevalence in % (References)	
				Serology	Feces
**The Americas**	**USA**	Pigs	GII.18, GII.11, GII.19	71 [46]	0–19 [25,28,46,48]
		Cattle	GIII.1, GIII.2	100 [49]	29–72 [50,51,52]
		Cats	GIV.2		17–43 [53,54]
		Sea lion	GII/GIV		9 [55]
	**Canada**	Pigs	GII, GII.11, GII.18		2–85 [30,31,32]
		Cattle	GIII.2		1 [30]
	**Venezuela**	Pigs	all		0 [39]
		Cattle	GIII		0.7 [56]
	**Argentina**	Cattle	GIII.1, GIII.2		3 [57]
	**Brazil**	Pigs	GII.11, GII.18, GII.19		0–52 [44,58,59,60,61]
		Cats	GIV,2		3 [62]
**Asia/New Zealand**	**China**	Pigs	GII.11, GII.18, GII.19		0–17 [26,27,33,35]
		Cattle	GIII.1		11 [63]
		Bats	NA		3–4 [6,64]
	**Taiwan**	Pigs	GII.11		1.6 [34]
	**Japan**	Pigs	GII.11	36 [46]	0.4–15 [10,36,45]
		Dogs	GIV		2 [65]
		Cats	GIV.2		1.2 [65]
		Rodents	GV		0–14 [66]
	**Korea**	Pigs	GII.11, GII.18		0.5–2 [37,67]
		Dogs	Canine norovirus	16 [68]	3 [68]
		Cattle	GIII.1, GIII.2		9 [69]
	**Iran**	Cattle	GIII.1, GIII.2		18–40 [70,71]
	**Turkey**	Cattle	GIII.2		4–9 [72,73]
	**India**	Cattle	GIII.1		0.4 [74]
	**New Zealand**	Pigs	GII.11		9 [38]
		Cattle	GIII.1		54 [75]
		Sheep	GIII.3		24 [38]
**Europe**	**Italy**	Pigs	GII.11, GII.18, GII.19		0–0.5 [76,77]
		Cattle	GIII.1, GIII.2		11–21 [78,79]
		Dogs	GIV, GVI	5–60 [47,80,81]	2–5 [82,83]
		Lion	GIV.2		100 [84]
		Cats	GIV.2	16 [85]	3 [81]
	**Spain**	Pigs	all		12 [86]
		Dogs	GVI		8 [83]
	**Portugal**	Dogs	GIV, GVI	64 [47]	23–28 [87,88,89]
	**Greece**	Dogs	GIV.2		8 [90]
	**France**	Cattle	GIII.1, GIII.2		20–37 [91,92]
		Dogs	GVI.2	20 [47,83]	0 [83]
	**Switzerland**	Dogs	GVI.2	20 [47]	
	**Germany**	Pigs	GII.18		14 [41]
		Cattle	GIII.1, GIII.2	66–99 [93,94]	93 [95]
		Dogs	GIV, GVI.2	16 [47]	4 [83]
		Rodents	GV		10 [96]
	**Netherlands**	Pigs	GII.11		2 [9]
		Cattle	GIII.2	0–44 [9]	4 [97]
		Dogs	GVI.2	34 [47]	
		Porpoise	not classified yet	24 [98]	10 [98]
	**Belgium**	Pigs	GII.19		4.6 [99]
		Cattle	GIII.2	93 [100]	4–9 [80,100,101,102]
	**UK**	Cattle	GIII.1, GIII.2	66–98 [93,103]	11 [104]
		Dogs	GIV, GVI, GVII	45–48 [47,105,106]	0 [106]
		Rodents	GV		22–67 [107]
	**Ireland**	Pigs	none		0 [40]
		Dogs	none	0 [47]	
	**Denmark**	Dogs	GVI.2	20 [47]	
		Rodents	none		0 [108]
	**Sweden**	Dogs	GVI.2	40 [47]	
	**Norway**	Cattle	GIII.1, GIII.2		50 [109]
		Dogs	GVI.2	32 [47]	
	**Finland**	Dogs	GVI.2	70 [47]	0 [110]
		Rodents	none		0 [111]
	**Poland**	Dogs	GIV.2	32 [47]	
	**Slovenia**	Pigs	GII.11, GII.18		1.2 [42]
		Cattle	GIII.2		2 [42]
	**Hungary**	Pigs	GII.11		6 [112]
		Dogs	GVI	0 [47]	3 [113]
		Rodents	GV		24–67 [114]
**Africa**	**Egypt**	Cattle	GIII.2		24 [115]
	**Tunisia**	Cattle	GIII.2		17 [116]
	**South Africa**	Pigs	none		0 [117]
	**Ethiopia**	Pigs	GII.1		0 [43]

**Table 2 viruses-11-00478-t002:** Summary of animal models for human norovirus.

	Gnotobiotic Calf [123]	Gnotobiotic Pig [151,152,153,154,155,156,157,158,159,160]	Mini Piglet [145]	Rhesus Macaque [136,142,161]	Pigtail Macaque [162]	Chimpanzee [163,164]	Balb/c RAG/γc^−/−^ Mouse [144]
Virus	GII.4	GII.4, GII.12	GII.3	GI.1, GII.2, GII.4, GII.17	GII.3	GI.1	GII.4
Inoculation (route and virus quantity)	Oral 1.62 × 10^7^ genomes	Oral/intranasal 10^4^–10^10^ genomes	Intragastric 10^7^ genomes	Oral/intragastric 10^5^–10^6^ genomes	Nasogastric, Quantity not clear	Intravenous/intragastric 4 × 10^6^–4 × 10^8^ genomes	Intraperitoneal 4 × 10^3^–7 × 10^4^ genomes
Shedding	3 days	2–16 days	7 days	1–19 days	Up to 21 days	2 days–17 weeks	No shedding ^1^
Seroconversion	Yes	Yes	NA	Yes/no ^2^	Yes	Yes	No
Pathology	Lesions, mild villous atrophy, enterocyte vacuolization in small intestine	Increase in inflammatory cells in LM, necrosis, shortening of villous tips	No damage	No damage	NA	No damage	No damage
Tropism (detection of viral antigen or genome)	Positive enterocytes in the ileum, cells in LM	Enterocytes and cells in LM of duodenum, jejunum, ileum. Spleen and MLN	Immune cells in the small/large intestine, tonsils, spleen, lymph nodes, MLN	NA	NA	Cells in LM of duodenum and jejunum	Macrophage-like cells in liver and spleen. Viral genomes detected in various tissue ^3^
Disease	Diarrhea	Diarrhea	Diarrhea	Asymptomatic	Diarrhea	Asymptomatic	Asymptomatic
Viremia	Yes (low)	Yes	Yes	NA	NA	NA	NA

^1^ When inoculated orally and intraperitoneal simultaneously, virus was shedded in feces. ^2^ Depending on study. ^3^ Stomach, small/large intestine, MSN, liver, spleen, kidney, heart lung, bone marrow. MSN = mesenteric lymph nodes, LM = lamina propria.

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
