# Peer review of "Animals as Reservoir for Human Norovirus"

_viruses, 2019, doi:10.3390/v11050478_

Round 1

Reviewer 1 Report

Infection of humans with norovirus (NoV) occurs annually with each 3-4 years the emergence of a pandemic strain that sweeps across the planet. Much like influenza virus NoV undergoes both antigenic drift and shift in maintaining this seasonal vs pandemic profile. But how is NoV maintained within the environment and what are the driving factors in producing the pandemic strains. Taking the influenza similarities into consideration it is then possible that human NoV may persist/infect animal reservoirs thus enabling its maintenance in the environment and its potential to recombine its genome. This article from Villabruna and colleagues undertook a systematic review of the current literature to determine if there is evidence that human NoV can infect animal reservoirs and/or whether animal NoV can infect humans. This is actually a very difficult thing to prove either way due to the lack of effective and efficient cell culture models, small animal models and some difficulties with cross-reactive serology. It is also confounded by the highly sensitive molecular tools that can identify NoV genomes but are they are indicating infection. This review though provides a very good assessment of the current literature and the current state of play on the NoV field.

Points for consideration:

-       Did the authors find any evidence for cross-species transmission that does not involve humans? i.e. can dogs and cats for example transmit NoV between them? This would most likely occur in more rural settings with domesticated animals dogs, chickens, cattle etc

-       Although the authors provided some mention of this in the Discussion I wonder if it would be appropriate for them to have a separate section on the merits or disadvantages of the serology and molecular/sequencing approaches used in the current literature. Could they provide some opinion of what they see are the pros and cons of each and perhaps give some indication on a direction?

-       Section 2.2.1 was a little hard to follow and extract what information was being identified and discussed. Perhaps consider rewriting this section to provide a better clarity

-       Tropism of human NoV appears highly restrictive and I would be helpful if the authors included a separate section on virus cultivation systems and the difficulties inherent within them. Can researchers infect dog cells with human NoV once isolated from the infected animal? This type of observation is curious and it would be interesting to know if this has been attempted.

Minor points:

-       Different font used on line 71 on pg 2

-       In Table 1 New Zealand seems to be tacked on the end next to Africa. It would seem more appropriate to have this near Asia and include data from Australia (if available)

-       - pg 6 line 115 I think “pub” should be “cub”?

Author Response

Reviewer 1: Points for consideration:

#1: Did the authors find any evidence for cross-species transmission that does not involve humans? i.e. can dogs and cats for example transmit NoV between them? This would most likely occur in more rural settings with domesticated animals dogs, chickens, cattle etc

Answer: We did not find any studies describing a transmission between animal viruses. For clarification we added this in the text (lines 184-186).

#2: Although the authors provided some mention of this in the Discussion I wonder if it would be appropriate for them to have a separate section on the merits or disadvantages of the serology and molecular/sequencing approaches used in the current literature. Could they provide some opinion of what they see are the pros and cons of each and perhaps give some indication on a direction?

Answer: As the reviewer noted, the advantage and disadvantages of serology and sequencing were already discussed in the review but we added some extra discussion points to the existing discussion (lines 405-410)

#3: Section 2.3.1 was a little hard to follow and extract what information was being identified and discussed. Perhaps consider rewriting this section to provide a better clarity.

Answer: To make this section more clear we restructured into a first section (lines 199–210) in which the serology results are described and a second section in which cross-reactivity is discussed (lines 210-222).

#4:  Tropism of human NoV appears highly restrictive and I would be helpful if the authors included a separate section on virus cultivation systems and the difficulties inherent within them. Can researchers infect dog cells with human NoV once isolated from the infected animal? This type of observation is curious and it would be interesting to know if this has been attempted.

Answer: For human norovirus two cell culture system have been described, although these systems only recently became available (Jones et al. Science 2014, Ettayebi et al. Science 2016). Murine norovirus can be replicated to high titers in Raw cells. Some attempts have been made to culture human and animal noroviruses in cells derived from animals, however, none of these were successful. Although the topic is very interesting, we considered it out of scope of this review to add a whole section on the challenges encountered with norovirus culturing. However, we added a sentence about attempts to grow animal/human norovirus in animal cell culture into the text and referred to the papers describing the existing culturing systems for norovirus as well as the review discussing further attempts to do so (lines 323-324).

Minor points:

#5    Different font used on line 71 on pg 2

Answer: We have adapted the fonts

#6   In Table 1 New Zealand seems to be tacked on the end next to Africa. It would seem more -opriate to have this near Asia and include data from Australia (if available)

Answer: We have adapted Table 1, no data was available for Australia

 #7 pg 6 line 115 I think “pub” should be “cub”?

Answer: we have corrected the typo

Reviewer 2 Report

The current review "Animals as reservoir for human Noroviruses" provides a comprehensive review of the recent literature on norovirus interspecies transmission. The review is excellently written, well-researched, succinct and timely. The manuscript provides a well-balanced account of the evidence for and against norovirus inter-species transmission. This is really important as the question of zoonotic and anthroponotic transmission will affect how and which norovirus research is conducted in the future. 

A paragraph on animal calicivirus transmission to humans would fit well in this story

1) Is there a significant difference between animal noro- and caliciviruses

2) What about the transmission of animal caliciviruses to humans i.e. SMSLV

The role of HBGAs as key determinants for the host tropism is debatable since despite a general correlation there are many strain-specific exceptions, animal strains that bind HBGA and some human strains/isolates that do not bind.

Minor Comments:

Figure 4: what do the half-circular arrows represent, please include in the legend

288-312: Please refer to ABH antigens or ABO blood groups (not ABO antigens)

290: "other" is confusing in this context

291: not sure that attachment is a pre-requisite for susceptibility, it certainly has an impact and could improve the chance of infection  but pre-requisite may be a bit too strong 

292: is there really evidence that HBGAs are markers for susceptibility on a cellular level. Guix et al 2007 showed increased binding upon FUT2 expression but no impact on susceptibility. Ettayebi et al 2016 showed that susceptibility of GII.3 strains in the enteroid system were independent of the secretor status

297: "ABO synthesis begins with the addition of fucose to a HBGA glycoprotein or glycolipid precursor structures." better "ABO synthesis begins with the addition of fucose to a carbohydrate precursor on glycoprotein or glycolipid precursor structures."

300-302: individuals either have a functional or a non-functional gene the genetics does not differ in cell types only the expression- please rephrase

303: To my knowledge Lewis antigens refer to the complete carbohydrate moiety not just to the additional fucose, maybe rephrase

321: GII and GV noroviruses may also recognize sialic acid Han et al. 2014

323/324: alpha-Gal expressed in all mammals except Hominidae. This does not make sense to me, dogs (Caddy et al.) and other mammals express FUT2 and produce HBGA. An alpha-Gal (Galili epitope) plus FUT2 would lead to a B-antigen. Please check and provide a reference.

333: encode for a protein OR contain gene

333: you mean FUT1 and FUT2 genes

Author Response

Reviewer 2:

A paragraph on animal calicivirus transmission to humans would fit well in this story

Answer: The subject of animal caliciviruses is very interesting. However, since these members of the Caliciviridae considered outside of the scope of this review and were not included in the systematic search we would prefer to not include a paragraph about animal calicivirus transmission in this review, however we did add lines 45-47.

#1: Is there a significant difference between animal noro- and caliciviruses

Answer: Viruses of the different genera of the caliciviridae family show at least 60% aa divergence. There are also differences in pathology with symptoms including haemorrhagic fever and sever respiratory problems. Many animal caliciviruses such as rabbit haemorrhagic disease virus (Lagovirus) as well as feline calicivirus (Vesivirus) have therefore a much higher mortality rate than animal norovirus which until now is understood to causing gastrointestinal problems only.

#2) What about the transmission of animal caliciviruses to humans i.e. SMSLV

Answer: Our initial literature search included caliciviruses but we did not come across many publications dealing with the zoonotic potential of non-norovirus caliciviruses. We have included an extra sentence in the introduction referring to some case reports about zoonotic and interspecies transmission cases referring to a paper that reviews these cases (Smith et al 1998) (lines 45–47).

The role of HBGAs as key determinants for the host tropism is debatable since despite a general correlation there are many strain-specific exceptions, animal strains that bind HBGA and some human strains/isolates that do not bind.

Minor Comments:

#3: Figure 4: what do the half-circular arrows represent, please include in the legend

Answer: An extra sentence was added to the figure (line 354-355)

#4: 288-312: Please refer to ABH antigens or ABO blood groups (not ABO antigens)

Answer: “ABO” was replaced with “ABH” in lines 343-344

#5: 290: "other" is confusing in this context

Answer: “Another” was removed from the sentence (Line 297)

#6: 291: not sure that attachment is a pre-requisite for susceptibility, it certainly has an impact and could improve the chance of infection but pre-requisite may be a bit too strong.

Answer: With this sentence we wanted to make clear that in order for a virus to enter the cell it is dependent on attaching/binding to the cell first. We are however aware that the cell needs to be both susceptible and permissive to support viral replication.

#7: 292: is there really evidence that HBGAs are markers for susceptibility on a cellular level. Guix et al 2007 showed increased binding upon FUT2 expression but no impact on susceptibility. Ettayebi et al 2016 showed that susceptibility of GII.3 strains in the enteroid system were independent of the secretor status.

Answer: As pointed out by the reviewer the exact role of HBGAs for norovirus infection is not fully understood, it is, however, the best understood/studied host factor to date (line 326-327). Also in non-secretors glycans are expressed that can be bound by norovirus, they are just lacking the fucose group. We have included an extra sentence addressing the alternative attachment factors that have been proposed with the according references (Tamura et al. 2004, Han et al 2014, Rydell et al 2009, Almand et al 2017) (lines, 328-329). We agree that the link between secretor status and host susceptibility is best studied for the  GII.4 genotype. This was in the above mentioned publication by by Ettayebi et al 2016 in which they show that secretor status was important for GII.4 infection in vitro but not for GII.3. In longitudinal studies this link has also been found (thorne et al 2018). It should also be noted that not many in vitro studies using cell culture have been published yet.

#8: 297: "ABO synthesis begins with the addition of fucose to a HBGA glycoprotein or glycolipid precursor structures." better "ABO synthesis begins with the addition of fucose to a carbohydrate precursor on glycoprotein or glycolipid precursor structures."

Answer: The sentence was replace with the proposed alternative sentence (lines 334-336).

#9: 300-302: individuals either have a functional or a non-functional gene the genetics does not differ in cell types only the expression- please rephrase.

Answer: The sentence has been changed accordingly (line 338-340)

#10: 303: To my knowledge Lewis antigens refer to the complete carbohydrate moiety not just to the additional fucose, maybe rephrase

Answer: As noted by the reviewer Lewis antigens do not refer to one fucose group and we have changed the sentence accordingly (lines 341-342)

#11: 321: GII and GV noroviruses may also recognize sialic acid Han et al. 2014

Answer: The attachment of GII and GV to sialic acids was mentioned in the review (line 329 and 369 respectively). The reference was added (line 329).

#12: 323/324: alpha-Gal expressed in all mammals except Hominidae. This does not make sense to me, dogs (Caddy et al.) and other mammals express FUT2 and produce HBGA. An alpha-Gal (Galili epitope) plus FUT2 would lead to a B-antigen. Please check and provide a reference.

Answer: The paper that states the lack of the alpha-Gal epitope due to inactivation of the alpga1,3galactosyltransferase (GGTA1 gene) in Hominidae is referenced in the review (ref 175, Zakhour et al 2009). The glycan they refer to is: Gala3Galb4GlcNAcb-R1

#13: 333: encode for a protein OR contain gene

Answer: The typo has been corrected to “…encode for a FUT1 and FUT2 protein”(Line 372)

#14: 333: you mean FUT1 and FUT2 genes

Answer: The typo in Lines 339 and 372 has been corrected to FUT1 and FUT2

Reviewer 3 Report

This review manuscript systemically reviewed the current knowledge regarding “Animals as reservoir for human Norovirus”. I think this is important and interesting topic. This information is helpful to the norovirus field. Overall, the authors did a very good job. However, there are a few points need to be addressed before it can be published.

1.       The authors should make clearly for the name of norovirus throughout the manuscript. The term “norovirus” is too board. Please specify the name of each specie and use human norovirus, porcine norovirus, murine norovirus, or bovine norovirus. For an example, lines 71-73, human norovirus or porcine norovirus? There are many similar scenarios.  This needs to be checked carefully.

2.       Table 1, why the results from references 25 and 26 were not included?

3.       Lines 165-188, is norovirus zoonotic or not? Please make it clear to the readers.

4.       Similarly, lines 188-211, make it clear whether human norovirus can be transmitted from humans to animals? From a pathogen can be transmitted from human to animal, it is called a “reverse zoonotic pathogen”.

5.       The authors should look at the publications regarding deep sequencing of stool samples of human or animal species. Can multiple genotypes or genogroups of norovirus be detected in stool samples from individual human or animal? Any viral quasispecies detected or not?

6.       There are some evidence suggest that recombination may occur between noroviruses of human and animals. The authors should include this topic and comment it.

Author Response

Reviewer 3: Major comments

#1: The authors should make clearly for the name of norovirus throughout the manuscript. The term “norovirus” is too board. Please specify the name of each specie and use human norovirus, porcine norovirus, murine norovirus, or bovine norovirus. For an example, lines 71-73, human norovirus or porcine norovirus? There are many similar scenarios.  This needs to be checked carefully.

Answer: The reviewer raised a valid point and we have specified the norovirus type throughout the manuscript where it was not clear.

#2:  Table 1, why the results from references 25 and 26 were not included?

Answer: They were included in Table 1 (see faeces prevalence section for pigs in China). The order of references is not chronological in the table since they were used perviously in the text.

#3: Lines 165-188, is norovirus zoonotic or not? Please make it clear to the readers.

#4:  Similarly, lines 188-211, make it clear whether human norovirus can be transmitted from humans to animals? From a pathogen can be transmitted from human to animal, it is called a “reverse zoonotic pathogen”.

Answer to 3 and 4: We have replaced “anthroponosis” with “reverse zoonosis” (lines 19, 387, 433). The question whether norovirus are zoonotic or reverse zoonotic is in our opinion still open. The aim of this review was to summarize the existing data on this topic and even though several cases have described evidence for transmission of human norovirus to animals, we consider the evidence not sufficient to conclude from it yet that norovirus cross the species barrier (line 425-427).

#5:  The authors should look at the publications regarding deep sequencing of stool samples of human or animal species. Can multiple genotypes or genogroups of norovirus be detected in stool samples from individual human or animal? Any viral quasispecies detected or not?

Answer: In our primary literature search publications that used NGS to look in human or animal faeces have been included. During the data extraction, papers were only included when they either described norovirus surveillance in animals or surveillance in humans with a focus on animal noroviruses. To our understanding only one study has found two different human norovirus genotypes in animals and we mention this in the text (lines 258-259). No study has reported the detection of animal norovirus sequences in human stool and therefore evidence is purely based on serology data which is discussed in section 2.3.1. Studies that describe diversity of human norovirus in human stool were not included since this was outside of the scope of this review.

#6:  There are some evidence suggest that recombination may occur between noroviruses of human and animals. The authors should include this topic and comment it.

Answer: To our knowledge only one human-animal recombinant has been described in the literature, namely GII(NA)-GVI, which is mentioned in the discussion together with the other intergenogroup recombinants (line 422-424). One other paper by Chhabra et al. virus research 2010 states to have found a recombinant between GII.b-GII.18, however what is stated in the paper as closest reference strain is not a GII.18 but a GII.21, which is a human norovirus genotype. This paper was therefore not included in the discussion.